# Cytochalasins from the Ash Endophytic Fungus *Nemania diffusa* DSM 116299

**DOI:** 10.3390/molecules30040957

**Published:** 2025-02-19

**Authors:** Özge Demir, Katharina Schmidt, Barbara Schulz, Theresia E. B. Stradal, Frank Surup

**Affiliations:** 1Department of Microbial Drugs, Helmholtz Centre for Infection Research, 38124 Braunschweig, Germany; oezge.demir@helmholtz-hzi.de; 2Institute of Microbiology, Technical University of Braunschweig, 38106 Braunschweig, Germany; b.schulz@tu-braunschweig.de; 3Department of Cell Biology, Helmholtz Centre for Infection Research, 38124 Braunschweig, Germany; katharina.schmidt@helmholtz-hzi.de (K.S.); theresia.stradal@helmholtz-hzi.de (T.E.B.S.)

**Keywords:** fungal endophytes, secondary metabolites, structure elucidation, cytotoxicity, actin inhibitors

## Abstract

The secondary metabolome of *Nemania diffusa*, isolated as an ash endophytic fungus, was analyzed in detail. From its cultures, a previously undescribed cytochalasin **1** was isolated using preparative HPLC, together with six known congeners: 18-dehydroxy-cytochalasin E (**2**), cytochalasins Z_7_ (**3**), Z_8_ (**4**), and E (**5**), 18-dehydroxy-17-didehydro-cytochalasin E (**6**), and K Steyn (**7**). The structures of these compounds were determined using data from high-resolution mass spectrometry (HR-MS), in combination with 1D and 2D nuclear magnetic resonance (NMR) spectroscopy. Metabolites **1**–**4** share a characteristic 12-membered lactone moiety, placing them within a rarely examined cytochalasin subclass. Thus, the compounds were incorporated into our ongoing screening campaign to study the structure–activity relationship of this metabolite family. We initially determined their cytotoxicity in eukaryotic mouse fibroblast L929 cells using an MTT-based colorimetric assay, and further investigated their effect on the cellular actin dynamics of the human osteosarcoma cell line U-2OS in detail. Unexpectedly, we discovered a high number of irreversible compounds (**1**, **2**, and **4**). Additionally, we highlighted specific structural features within the 12-membered cytochalasin subclass that may play a role in directing the reversibility of these compounds.

## 1. Introduction

All types of plants harbor endophytes, endosymbiotic microorganisms—mostly bacteria and fungi—that are living inside plant tissues without causing harm to their host [1,2]. Endophytic fungi can even play a crucial role in supporting plant health, in particular under biotic and abiotic stress conditions. However, this term refers only to the moment of detection, without regard for the future status of the interaction. Some endophytes can switch to a parasitic lifestyle, considered to be latent pathogens.

The association between endophytic fungi and their host plants represents a complex interkingdom interaction, which includes modulating the host plant’s defense mechanisms, regulating fungal virulence factors, and engaging in both antagonistic and cooperative interactions with other members of the microbiome. As a result of these interactions, endophytic fungi produce a wide range of secondary metabolites that have potential biotechnological applications [3].

As long as the delicate equilibrium between host plant defense and fungal virulence is maintained, the plant is healthy and colonized by the fungal endophyte. A stable equilibrium between all members of the microbiome and the host ensures plant health [4]. Consequently, our ongoing project has the long-term goal of finding endophyte(s) that contribute to protect the host against the invasive fungal pathogen of ash, *Hymenoscyphus fraxineus*. For this purpose, we are exploring the fungal metabolites of candidate ash endophytes.

In this study, we describe metabolites from *Nemania diffusa* DSM 116299 (syn. *Hypoxylon vestitum)*, isolated from twigs of *Fraxinus excelsior* growing in the Elm forest near Erkerode, Lower Saxony, Germany. The genus *Nemania* has been recognized as the source of multiple secondary metabolites in the past; cytochalasins have previously been isolated from the strain *Nemania* sp. UM10M [5]. Other recent examples are the isolation of guanacastane-type diterpenoids from the endolichenic fungus *Nemania* sp. EL006872 [6], cytotoxic tropolones, and isochromenones from *Nemania* sp. BCC 30850 [7], as well as rare furanones from *Nemania serpens*, isolated as an endophyte from Riesling grapevines [8].

Chemical investigation of this strain led to the isolation and structure elucidation of a new cytochalasin **1**, along with the identification of six previously known cytochalasins (**2**–**7**). This study provides detailed insights into their isolation, structural characterization, and cytotoxic activities in mouse fibroblast L929 cells, as well as their disruptive effects on the F-actin network in the human osteosarcoma cell line U-2OS.

## 2. Results and Discussion

### 2.1. Isolation and Structure Elucidation of the Compounds

The ethyl acetate (EtOAc) extract of *N. diffusa* obtained from cultures grown on malt extract agar medium was purified using preparative HPLC, yielding metabolites **1**–**7** (Figure 1). The known compounds 18-dehydroxy-cytochalasin E (**2**), cytochalasins Z_7_ (**3**), Z_8_ (**4**), and E (**5**), 18-dehydroxy-17-didehydro-cytochalasin E (**6**), and K Steyn (**7**) were identified by comparing their HRESIMS and NMR spectroscopic data with those reported in the literature [9,10,11,12,13].

The new metabolite **1** was isolated as a brownish oil. Its molecular formula, C_28_H_35_NO_6_, implying twelve degrees of unsaturation, was deduced using the molecular ion peak cluster at *m/z* 482.2526 [M + H]^+^ in the HR-ESI-MS spectrum of **1** (calculated for C_28_H_36_NO_6_ 482.2537).

The ^1^H NMR and HSQC NMR spectra of **1** showed the presence of seven aromatic/olefinic signals (two with dual intensities), as well as seven aliphatic methines, one methylene, and four methyl groups (Table 1). The ^13^C NMR data revealed the further presence of two carboxylic carbons and one aromatic and three oxygen-binding aliphatic carbons without bound protons. Four spin systems were assembled using COSY correlations: the phenyl ring of H–2′ to H–6′, 10–H_2_/3–H/4–H/5–H/11–H_3_, the large system 7–H/8–H/13–H/14–H/15–H_2_/16–H(22–H_3_)/17–H/23–H_3_, and the isolated double bond 19-H/20-H (Figure 2A). These were connected by HMBC correlations, revealing the tricyclic cytochalasin core structure. Most significant were the HMBC correlations from H–3 and H–4 to C–1 and C–9, from H–8 to C–9, and from H–19 and H–20 to C–21 in this process. The number of unsaturations requires a connection between C–9 and C–21.

The stereochemistry of **1** was assigned via the comparison of the chemical shifts of **1** to the known congeners. The ^13^C data of carbons inside the central six-membered ring structure were highly similar to those of **2** and **4**–**6**, in particular to those of the chiral centers C–5, C–6, and C–7, which may vary. Thus, we conclude a common configuration of **1**, **2**, **4**, **5,** and **6**. Strong ROESY correlations between H–15 and H–17 as well as H–17 and H–19 confirmed that these protons are located below the main molecular plane (Figure 2B). On the other hand, H–16, H–20, and H_3_–23 are pointing upwards, as confirmed by the ROESY correlations between H–16 and H_3_–23 and H–20 and H_3_–23. These findings indicate a 17*R*, 18*R* configuration.

The closest known structural relative of **1** is compound **2**, with **1** being the 18-hydroxy derivative of **2**. This metabolite was initially described as a planar structure with unassigned stereochemistry, isolated from an endophytic fungus in the genus *Rhinocladiella* [9]. Later, the compound was rediscovered in the marine-derived fungus *Spicaria elegans* and published with an assigned stereochemistry; however, no experimental data or explanation was provided [14]. Thus, we addressed this question by chemical shift analysis and the pattern of ROESY correlations [15]. Most notably, the correlation between H–17 and H–15b confirms the depicted configuration (Appendix A).

Furthermore, 18-dehydroxy-17-didehydro-cytochalasin E (**6**) was described in 2017 under the name aspochalasin R [12]. However, we prefer to avoid using this trivial name, as it was assigned in 2010 to an unrelated metabolite from the marine-derived fungus *Spicaria elegans* [16]. Additionally, **6** does not contain the leucine-derived moiety characteristic of other aspochalasins.

### 2.2. Biological Evaluation of Cytochalasins

Metabolites **1**–**4** are classified within the lactone cytochalasin subclass, a group that has not been extensively studied in the existing literature, especially in terms of their effects on the actin cytoskeleton. Actin plays a crucial role in various cellular processes, making it essential to understand how these compounds interact with it to advance our knowledge. In light of this gap in the research, we decided to investigate this set of compounds to delve deeper into the relationship between their chemical structure and biological activities.

Compounds **1**–**4** were tested for cytotoxic effects in a mouse fibroblast cell line L929 using an MTT colorimetric assay. The evaluated compounds demonstrated moderate to weak cytotoxicities, with concentrations ranging from 14 µM to 43 µM (see Table 2).

Based on the fact that cytochalasans are known as actin cytoskeleton inhibitors, we examined the isolated cytochalasins **1**–**4** in a well-established actin disruption assay, described by Kretz et al. [17]. Briefly, human osteosarcoma cells (U-2OS) were seeded on fibronectin-coated coverslips and treated with two different concentrations of **1**–**4**, corresponding to their previously determined IC_50_ values in L929 mouse fibroblasts—1× IC_50_ corresponds to a low-dose concentration (LD) and 5× IC_50_ corresponds to a high-dose concentration (HD). Filamentous actin (F-actin) was stained using ATTO488-coupled phalloidin to visualize the effects of the compounds on the actin network (see Figure 3). Compounds **1**–**4** belong to the so-called lactone cytochalasin subclass (cytochalasins bearing a lactone functional group in the macrocycle [18]) virtually uncharacterized in the literature regarding their effects on actin. Accordingly, we used this set of compounds to enhance our understanding of the relationship between the structure and biological activity of cytochalasins.

In the case of **1**, the actin cytoskeleton of U-2OS cells was strongly affected, starting with the depletion of F-actin-containing structures, such as lamellipodia—protrusive actin meshwork at the cell periphery (green arrowheads in Figure 3d,m,o), stress fibers—antiparallel, contractile, cable-like structures (yellow arrowheads in Figure 3d,m,n,o), and the formation of large actin accumulations after LD treatment (red arrowheads in Figure 3a). The increased concentration of **1** resulted in the complete collapse of the actin network (Figure 3b), which was irreversible despite wash out experiments (Figure 3c). Similar disruption events were observed for **5** and **7** (see Appendix A), consistent with the results recently published by Pourmoghaddam et al. [19].

In addition to **1**, the effects of **2** on actin were less pronounced after LD treatment, although this cytochalasin only lacks hydroxylation at C-18 found in the backbone of **4**. LD-treated cells still exhibited lamellipodia and stress fibers comparable to the DMSO control (Figure 3m–o). However, HD application of **2** clearly caused characteristic actin accumulations (Figure 3e). Notably, the HD effects of **1** and **2** were not reversible after a one-hour recovery time (Figure 3c,f).

Next, we observed the weak LD but distinct HD effects of compound **4** (Figure 3j,k), similar to **2**. In contrast, compound **3**, which contains a methylidene group within the six-membered ring instead of a double bond, resulted in a slight reduction in stress fibers and a notable retraction of lamellipodia after LD treatment (Figure 3g). Increasing the concentration of **3** led to a complete collapse of the actin network (Figure 3h). Surprisingly, **3** is the sole compound showing significant but still incomplete recovery of the actin network upon a one-hour recovery period in a medium lacking the compound (compare Figure 3i,l).

In this study, we present the first set of 12-membered cytochalasins screened for bioactivity, in particular their effects on the actin cytoskeleton. All compounds exhibited moderate cytotoxic effects on L929 mouse fibroblasts and distinct activities on actin dynamics. Strikingly, we discovered a remarkably high number of irreversible compounds (**1**, **2**, and **4**), although only a few cytochalasans have been assigned this feature in the literature—for example, pseudofuscochalasin A [20], cytochalasin E [19] (this study, see Appendix A), chaetoglobosin D, and deoxaphomin [17]. Importantly, none of them belong to the group of 12-membered cytochalasins.

The primary structural differences between the herein-reported reversible (e.g., **3**) and irreversible (e.g., **2** and **4**) cytochalasins are located within the six-membered ring and concern (i) the arrangement of the double bond and (ii) the presence of an epoxy group. This observation leads us to hypothesize that specific chemical modifications within the six-membered ring, especially the existence of a methylidene group, may favor the reversibility in twelve-membered cytochalasins. Recently, we also noted this trend when comparing the reversibility of deoxaphomin (irreversible) and deoxaphomin B (reversible), both of which belong to the 13-membered cytochalasins [21].

It is worth noting that trace amounts of impurities were detected in some samples (mainly **1** and **3**) using NMR and LC-MS; in particular, **3** contains unidentified cytochalasin congeners. Due to the limited quantities of these compounds, repeated purification was not feasible, making it difficult to completely rule out the possibility of cross-reactions with other unidentified cytochalasins. Therefore, we have to interpret these data with caution. Nevertheless, the initial screening of these 12-membered cytochalasins on actin highlighted their remarkable behavior regarding their activity and reversibility on the actin cytoskeleton, paving the way for further studies on this unique subclass of cytochalasins.

### 2.3. Cytochalasin Production of Nemania spp.

Cytochalasins are well known as phytotoxic agents. In particular, cytochalasin E (**5**) has been shown to be strongly phytotoxic, i.e., exhibiting phytotoxic effects on *Lactuca sativa * and *Raphanus sativus* L. seedlings, with effects greater than those of the positive control, glyphosate [22]. It is also strongly anti-fungal, i.e., against *Paracoccidioides brasiliensis* with a minimal inhibitory concentration of 3.6 µM and minimum fungicidal concentration of 7.2 µM [23].

Many cytochalasans have been isolated from plant-associated fungi, including both pathogens and endophytes, with their biological roles often remaining unclear. A number of phytotoxic cytochalasins have been found in fungal plant pathogens, for instance the cytochalasins B, F, Z2, and Z3 [22].

The production of cytochalasins by *Nemania diffusa* DSM 116299 could serve as a virulence factor, enabling it to be a latent pathogen when the host’s defense is weakened. However, these metabolites may also play a role in maintaining equilibrium in its interactions with other members of the microbiome, namely, it is known to be a mycoparasite [23]. Members of the cytochalasin family have previously been isolated from the genus *Nemania,* namely, *Nemania* sp. UM10M, which had been isolated from a diseased *Torreya taxifolia* leaf and produced 19,20-epoxycytochalasins C and D, as well as 18-deoxy-19,20-epoxy-cytochalasin C [9].

## 3. Materials and Methods

### 3.1. General Chemical Procedures

Optical rotations were determined in methanol (Uvasol, Merck, Darmstadt, Germany) using a PerkinElmer 241 polarimeter (Anton-Paar Opto Tec GmbH, Seelze, Germany), and UV spectra were recorded with a Shimadzu UV/Vis spectrophotometer UV-2450 (Kyoto, Japan).

Then, 1D and 2D nuclear magnetic resonance (NMR) spectra were obtained using a Bruker Avance III 500 MHz spectrometer (Billerica, MA, USA) fitted with a BBFO (Plus) SmartProbe (^1^H 500 MHz, ^13^C 125 MHz) and Bruker Avance III HD 700 MHz spectrometer (^1^H 700 MHz, ^13^C 175 MHz) equipped with a 5mm TCI cryoprobe. 

HPLC-ESIMS spectra were collected using an ion trap mass spectrometer (amaZon speed™, Bruker) operating in both negative and positive ionization modes, whereas HR-ESIMS spectra were obtained using a MaXis ESI-TOF mass spectrometer from Bruker. Both MS devices were connected to Agilent 1200 Series HPLC systems (Agilent Technologies Deutschland GmbH and Co. KG, Waldbronn, Germany), equipped with an Acquity UPLC BEH C_18_ column (2.1 × 50 mm, 1.7 µm) from Waters (Eschborn, Germany) as the stationary phase. The mobile phase consisted of solvent A (Milli-Q H_2_O (Millipore, Burlington, MA, USA) + 0.1% formic acid) and solvent B (acetonitrile + 0.1% formic acid) with the following gradient: 5% B for 0.5 min, increasing to 100% B in 19.5 min, and then maintaining isocratic conditions at 100% B for 5 min. The flow rate was set at 0.6 mL/min, and UV/Vis detection was performed at 200–600 nm.

All chemicals and solvents were acquired from AppliChem GmbH (Darmstadt, Germany), Avantor Performance Materials (Deventor, The Netherlands, Carl Roth GmbH and Co. KG (Karlsruhe, Germany), and Merck KGaA (Darmstadt, Germany) in analytical and HPLC grade.

### 3.2. Fungal Material

The source of DSM 116299 was a necrotic twig from *Fraxinus excelsior*, collected in the Elm Forest near Erkerode (52.204514, 10.719210). DSM 116299 was isolated by B. Schulz (TU Braunschweig) on 26 October 2020, under the internal strain code 10622.

The identification of *Nemania diffusa* was based on its morphology, mycoparasitism, and molecular analysis. In co-culture with the pathogen *H. fraxineus*, it grew towards the pathogen as an imperfect white mycelium, overgrowing it and lysing the hyphae, as seen microscopically (see also [24]).

### 3.3. Molecular Analysis

For molecular analysis, we have performed DNA extraction (MasterPureTM Yeast DNA purification kit, Jena Bioscience, Jena, Germany), followed by the amplification and sequencing of the rDNA ITS (internal transcribed spacer) fragment, which serves as the primary common DNA barcode for fungi [25]. For more accurate identification, we sequenced the partial genes of RPB2. The primers used for sequencing the rDNA ITS region and partial 28S large subunit (LSU) of the rDNA operon were ITS4 [26] and NL4 [27]. The primers used for the sequencing of the partial RPB2 gene were RPB2-5F and RPB2-7cR [28]. The comparison of assembled DNA sequences was performed with GenBank databases (https://blast.ncbi.nlm.nih.gov/Blast.cgi?PROGRAM=blastn&PAGE_TYPE=BlastSearch&LINK_LOC=blasthome (accessed on 16 January 2025)). The data can be obtained under NCBI GenBank accession number PQ686754 and the Appendix A, respectively.

### 3.4. Small-Scale Fermentation

The small-scale cultivation of the fungus was carried out in three different liquid media: Q6 ½ medium (10 g/L glycerin, 2.5 g/L D-glucose, 5 g/L cotton seed flour, and pH = 7.2); YM 6.3 medium (10g/L malt extract, 4g/L yeast extract, 4g/L D-glucose, and pH = 6.3); and ZM ½ medium (5 g/L molasses, 5 g/L oatmeal, 1.5 g/L D-glucose, 4 g/L saccharose, 4 g/L mannitol, 0.5 g/L edamine ammonium sulfate, 1.5 g/L calcium carbonate, and pH = 7.2). Cultivation was also performed on one solid medium (malt extract agar: 20 g/L malt extract, 0.1 g/L yeast extract, 12 g/L agar, and pH 5.6). A 7-day-old mycelial culture from a YM agar plate was cut into small pieces using a 7 mm cork borer.

For liquid cultures, five pieces were used to inoculate 500 mL Erlenmeyer flasks containing 200 mL of medium. The cultures were incubated at 23 °C on a rotary shaker (140 rpm). Fungal growth was monitored daily by measuring free glucose levels using Medi-Test glucose strips (Macherey Nagel, Düren, Germany). Fermentation was stopped 3–4 days after glucose depletion, and the biomass and supernatant were separated using vacuum filtration. Subsequently, the supernatants were extracted with an equal amount of EtOAc and filtered through anhydrous sodium sulfate. The EtOAc extracts were then evaporated to dryness at 40 °C using a rotary evaporator. The mycelia were extracted with 200 mL of acetone in an ultrasonic bath for 30 min at 40 °C, then filtered, and the filtrate was evaporated. The remaining aqueous phase was suspended in an equal volume of distilled water and subjected to the same procedure as described for the supernatants.

For solid agar cultures, well-grown mycelial cultures on YM agar plates were cut into small pieces using a 7 mm cork borer, and each agar plate was inoculated with a single piece on selected media. The cultures were then incubated in the dark at 23 °C for 3 weeks. For extraction, the agar cultures were cut into pieces and placed into a Schott bottle. The cultures were subsequently extracted with 100–200 mL of EtOAc for 30 min using a magnetic stirrer. Following filtration, the EtOAc was evaporated to dryness under reduced pressure at 39 °C.

### 3.5. Scale-Up Fermentation

The endophytic strain 10622 was cultivated on malt extract agar (MEA) medium. Ten well grown, 7-day-old YM agar plates of the mycelial culture were cut into small pieces using a 7 mm cork borer, and one piece was used to inoculate each of the fifty-five MEA plates. The cultures were then incubated in the dark at 23 °C for three weeks. Extraction was performed as detailed in Section 3.4 for solid agar cultures, resulting in the crude extract.

### 3.6. Isolation of the Compounds

Initially, the crude extract was filtered using an SPME Strata-X 33 µm polymeric reversed phase (RP) cartridge (Phenomenex, Aschaffenburg, Germany). The crude extract was then purified using a Gemini LC column (250 × 50, 10 µm; Phenomenex, Torrance, CA, USA). The mobile phase consisted of solvent A (H_2_O + 0.1% formic acid) and solvent B (MeCN + 0.1% formic acid), with a flow rate of 35 mL/min. The gradient was as follows: starting with 25% B for 3 min, increasing to 70% in 60 min, then to 100% B in 10 min, and maintaining 100% B for 10 min. This led to the isolation of **1** (0.99 mg, *t_R_*: 9.7 min), **5** (10.6 mg, *t_R_*: 9.7 min), and **6** (0.96 mg, *t_R_*: 11.2 min).

Fractions F23 and F24 were combined, since they contained the same compounds. The merged fractions were further purified using an Agilent Technologies 1200 Infinity Series semi-preparative HPLC, with an elution gradient starting at 55% B for 3 min, increasing to 70% in 15 min, then to 100% in 3 min, and maintaining 100% B for 2 min. A Water X-Bridge C18 column (5 µm; 250 × 10 mm) was used as the stationary phase. Compound **2** (0.7 mg) eluted at *t_R_* = 10.8 min.

Fraction F18 was further purified using the same semi-prep instrument, with a Gemini C18 column (5 µm; 250 × 10 mm) as the stationary phase. The gradient started at 50% B for 3 min, increased to 70% over 15 min, then to 100% over 3 min, and was maintained at 100% B for 2 min, yielding **7** (0.9 mg, *t_R_*: 9.4 min). Fractions F20 and F17 were further purified using the same instrument and column, with a gradient starting at 50% B for 3 min, increasing to 60% in 25 min, then to 100% 3 min, and maintaining 100% B for 2 min, yielding **3** (0.41 mg, *t_R_*: 9.6 min) and **4** (0.51 mg, *t_R_*: 10.2 min).

### 3.7. Spectral Data

Compound **1**: Light brown oil, ${\left[\alpha \right]}_{D}^{20}$ = + 8 (c = 0.001, MeOH); UV/Vis (0.005 mg/mL, MeOH) λmax (logε) 205 (4.5) nm; ^1^H NMR (700 MHz, CHCl_3_-*d*) and ^13^C NMR (175 MHz, CHCl_3_-*d*): see Table 1; and HR-ESIMS: *m/z* 482.2526 [M + H]^+^ (calculated for C_28_H_36_NO_6_, 482.2537).

Compound **2**: 18-dehydroxy-cytochalasin E: Colorless oil; HR-ESIMS: *m/z* 466.2583 [M + H]^+^ (calculated for C_28_H_36_NO_5_, 466.2588); data are in good agreement with those of [9].

Cytochalasin Z_7_ (**3**): White powder; HR-ESIMS: *m/z* 466.2586 [M + H]^+^ (calculated for C_28_H_36_NO_5_, 466.2588); data are in good agreement with those of [10].

Cytochalasin Z_8_ (**4**): White powder; HR-ESIMS: *m/z* 466.2584 [M + H]^+^ (calculated for C_28_H_36_NO_5_, 466.2588); data are in good agreement with those of [10].

Cytochalasin E (**5**): Colorless amorphous solid; ^1^H NMR (500 MHz, CHCl_3_-*d*): *δ*_H_ 7.33 (br t, *J* = 7.5 Hz, H–3′, and H–5′), 7.27 (br t, *J* = 7.5 Hz, H–3′, and H–5′), 7.15 (br d, *J* = 7.5 Hz, H–4′), 6.48 (d, *J* = 11.6 Hz, H–17), 5.89 (dd, *J* = 15.0, 8.8 Hz, H–10), 5.60 (d, *J* = 11.6 Hz, H–16), 5.22 (ddd, *J* = 15.0, 11.0, 3.8 Hz, H–11), 3.74 (m, H–3), 3.02 (dd, *J* = 5.0, 2.8 Hz, H–4′), 2.69 (dd, *J* = 13.7, 7.3 Hz, Ph-CH_2_), 2.65 (m, H–8), 2.65 (m, H–12a), 2.63 (m, H–7), 2.28 (qd, *J* = 7.3, 5.3 Hz, H–5), 2.15 (m, H–12b), 1.48 (s, H_3_–15Me), 1.25 (s, H_3_–6Me), 1.16 (d, *J* = 6.9 Hz, H_3_–13Me), and 1.10 (d, *J* = 7.3 Hz, H_3_–5Me) ppm; ^13^C NMR (125 MHz, CHCl_3_-*d*): *δ*_C_ 211.7 (C–14), 170.1 (C–1), 149.3 (C–19), 142.1 (C–17), 135.9 (C–1′), 131.4 (C–11), 129.6 (C–2′/C–6′), 128.9 (C–3′/C–5′), 128.4 (C–10), 127.3 (C–4′), 120.4 (C–16), 87.1 (C–9), 76.7 (C–15), 60.6 (C–7), 57.3 (C–6), 53.7 (C–3), 47.8 (C–4), 45.7 (C–8), 44.8 (PhCH_2_), 40.8 (C–13), 39.0 (C–12), 35.8 (C–5), 24.3 (15Me), 20.1 (13Me), 19.7 (6Me), and 13.1 (5Me) ppm; HRESIMS: *m/z* 496.2336 [M + H]^+^ (calculated for C_28_H_34_NO_7_ 496.2330), data are in good agreement with those of [11].

Compound **6**: White amorphous powder; HR-ESIMS: *m/z* 480.2392 [M + H]^+^ (calculated for C_28_H_34_NO_6_, 480.2381); data are in good agreement with those of [12].

Cytochalasin K Steyn (**7**): White amorphous powder; HR-ESIMS: *m/z* 496.2326 [M + H]^+^ (calculated for C_28_H_34_NO_7_, 496.2330); data are in good agreement with those of [13].

### 3.8. Cell Culture

The mammalian cell line L929 (mouse fibroblast, DSMZ ACC 2) was maintained in Dulbecco’s modified minimum essential medium (DMEM, Life Technologies, Carlsbad, CA, USA) and supplemented with 10% fetal bovine serum (FBS, Life Technologies, Carlsbad, CA, USA). The human osteosarcoma cell line (U-2OS, ATCC HTB-96) was maintained in complete growth medium containing DMEM (Life Technologies, Carlsbad, CA, USA) and supplemented with 10% FBS (Sigma-Aldrich, St. Louis, MO, USA), 1% L-glutamine (Life Technologies, Carlsbad, CA, USA), 1% sodium pyruvate (Life Technologies, Carlsbad, CA, USA), 1% minimum essential medium nonessential amino acids (MEM NEAA, Life Technologies, Carlsbad, CA, USA), and 1% penicillin–streptomycin (10,000 U/mL, Life Technologies, Carlsbad, CA, USA). Both cell lines were routinely cultured at 37 °C and in 7.5% CO_2_.

### 3.9. Cytotoxicity Assay

The cytotoxicity of **1**–**7** was determined by a colorimetric tetrazolium dye MTT assay, described in detail [29]. Briefly, 50,000/mL L929 cells were seeded onto a 96-well flat-bottom microtiter plate, followed by the addition of a serial dilution of the test compounds. The growth inhibition (IC_50_) was determined after 5 days of incubation. The compounds were dissolved in MeOH (1 mg/mL). MeOH was used as a negative control, and epothilone (1 mg/mL) was used as the positive control.

### 3.10. Actin Disruption Assay

To evaluate the biological effects of **1**–**7** on the actin cytoskeleton, we performed a 24 h endpoint assay followed by a staining procedure for filamentous actin (F-actin) as described in [17]. Briefly, 20,000 U-2OS cells were seeded on fibronectin-coated coverslips (25 µg/mL diluted in phosphate-buffered saline [PBS]) and allowed to spread overnight at 37 °C and in 7.5% CO_2_. The complete growth medium was pre-equilibrated at 37 °C and in 7.5% CO_2_ and was supplemented with **1–7** at concentrations corresponding to 1× IC_50_ (low dose: LD) and 5× IC_50_ (high dose: HD) [30]. Cells were treated with the pre-mixed medium for one hour and afterwards fixed using 4% paraformaldehyde in PBS for 20 min at 37 °C. Cells were permeabilized using 0.1% Triton X-100 in PBS for one minute at room temperature (RT). After three wash steps in PBS, the coverslips were incubated in PBS containing 1:100 ATTO488-phalloidin to stain actin filaments. After 1 h of incubation at RT, the coverslips were washed three times in PBS and mounted on objective slides using ProLong Diamond Antifade Mountant (Invitrogen, Carlsbad, CA, USA) containing DAPI for nuclear DNA staining.

To examine the reversibility of the tested compounds, we performed additional wash out steps using PBS after one-hour high-dose treatment, and allowed the cells to recover for one further hour in new growth medium prior to the aforementioned fixation and staining procedure.

The stained cells were viewed using an inverted microscope (Nikon eclipse Ti2, Tokyo, Japan) equipped with a 60× Nikon oil immersion objective (Plan Apofluar, 1.4 NA), a pco.edge back-illuminated sCMOS camera (Excelitas Technologies, Mississauga, ON, Canada), and a pE-4000 (CoolLED, Andover, UK) as a light source. The microscopy system was operated by and images were acquired with NIS (Nikon, Tokyo, Japan), and subsequently were processed with Image J v1.53t (NIH, Bethesda, MD, USA).

## 4. Conclusions

The present study revealed seven cytochalasins synthesized by *Nemania diffusa* DSM 116299, an endophytic fungus isolated from ash (*Fraxinus excelsior*). This represents the first report of cytochalasins being produced by an ash-associated endophytic fungus. Structural elucidation of the metabolites led to the discovery of one novel compound, alongside six previously known congeners. Notably, cytochalasins **1**–**4**, recognizable by their characteristic 12-membered lactone ring moiety, have been exposed as irreversible actin modulators, underscoring their potential for unique biological activity.

## Figures and Tables

**Figure 1 molecules-30-00957-f001:**
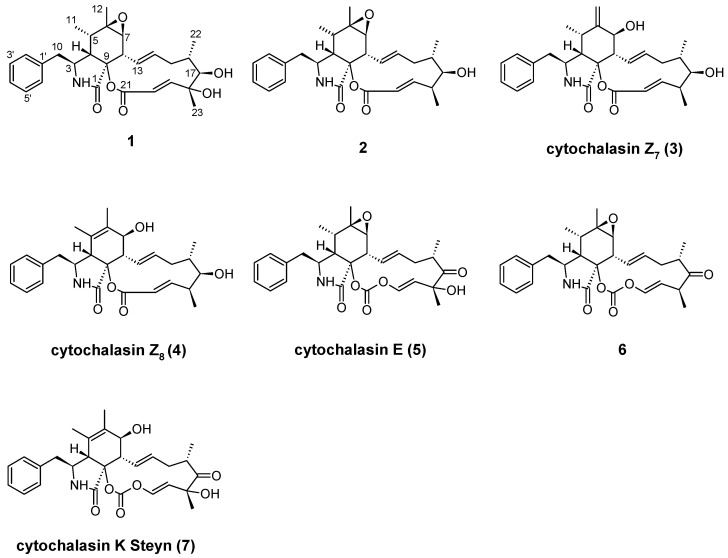
Chemical structures of cytochalasins **1**–**7** isolated from *Nemania diffusa* DSM 116299.

**Figure 2 molecules-30-00957-f002:**
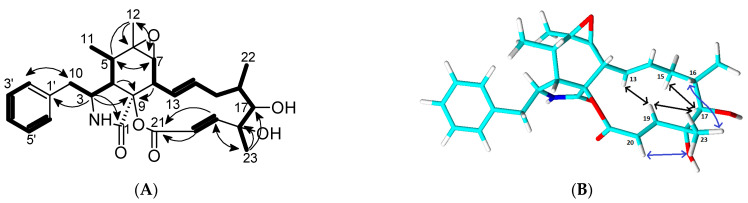
(**A**) Key COSY (bold bonds) and HMBC (arrows) correlations for **1**; (**B**) key ROESY correlations above (blue arrows) and below (black arrows) the molecular plane.

**Figure 3 molecules-30-00957-f003:**
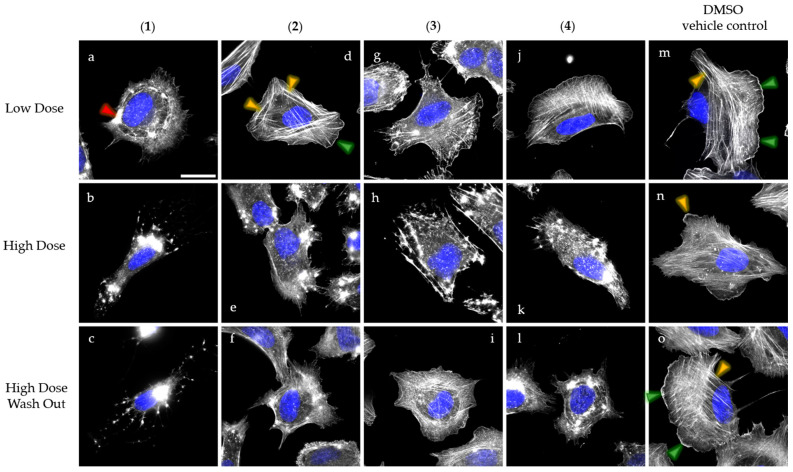
Overlay images of pseudocolored U-2OS cells treated with low dose (**upper row**) and high dose (**middle row**) concentrations of the herein-isolated cytochalasins **1** (**a**,**b**), **2** (**d**,**e**), **3** (**g**,**h**), and **4** (**j**,**k**). Wash out experiments to test for reversibility of high-dose effects are presented in the lower row (**c**,**f**,**i**,**l**,**o**). DMSO served as negative control (**m**–**o**). Concentrations for treatment were calculated based on IC_50_ values previously determined in mouse fibroblast L929 cells (low dose: 1× IC_50_; high dose 5× IC_50_). Cells were fixed after 1h treatment using paraformaldehyde and stained for F-actin using fluorescently coupled ATTO488-phalloidin (greyscale) and for nuclear DNA using DAPI (pseudocolored in blue). F-actin containing structures such as lamellipodien (green arrowheads) and stress fibers (orange arrowheads) were highlighted in (**d**,**m**–**o**). F-actin rich accumulations caused by cytochalasin treatment are exemplary, as marked by a red arrowhead in (**a**). Representative scale bar in (**a**) corresponds to 25 µm.

**Table 1 molecules-30-00957-t001:** NMR spectroscopic data (^1^H 700 MHz, ^13^C 175 MHz) of **1** in CDCl_3_.

Pos.	*δ_C_*_,_ Type	*δ_H_*, Mult. (*J* in Hz)	COSY	HMBC	ROESY ^a^
1	172.6, C				
3	51.0, CH	3.69, m	4, 10	1, 4, 5 > 9	11, 12, 10, 2′/6′, 4
4	49.0, CH	3.07, d (6.0)	3, 5	1, 5, 6, 9, 10	5, 10, 11, 3
5	35.9, CH	2.27, tq (6.9, 6.0)	4, 11	3, 4, 6, 11, 12	4, 11
6	57.1, C				
7	60.8, CH	2.77, d (5.5)	8	6, 8, 12, 13	12, 8, 13
8	47.0, CH	3.04, dd (10.0, 5.5)	7, 13	7, 9, 13, 14	5, 14
9	85.5, C				
10	45.0, CH_2_	2.86, dd (13.0, 8.6)	3, 10b	3, 4, 1′, 2′/6′	10b, 4, 3, 2′/6′
		2.80 (dd, 13.0, 6.5)	3, 10a	3, 1′, 2′/6′	10a, 4, 3, 2′/6′
11	12.6, CH_3_	0.99, d (6.9)	5	4, 5, 6	3, 12, 5, 4
12	19.4, CH_3_	1,18, s		5, 6, 7	11, 7, 3
13	124.5, CH	6.07, dd (15.2, 10.0)	8, 14	15	15, 7, 14
14	138.4, CH	5.20 (m)	13, 15	8, 15, 16	8, 15, 16, 13, 19
15	42.8, CH_2_	2.10 (m)	14, 16	13, 14, 16, 17, 22	22, 16, 14, 13
16	33.3, CH	1.54 (m)	15, 17, 22		23, 22, 15, 14
17	81.6, CH	3.42 (m)	16, 23	15, 18, 28	15, 19, 16, 22, 23
18	75.5, C				
19	154.7, CH	6.88, d (15.9)	20	18, 20, 21, 23	17, 16, 20, 23, 14
20	122.8, CH	5.97, d (15.9)	19	18, 21	23, 19, 17
21	166.8, C				
22	17.5, CH_3_	1.02, d (7.3)	16	15, 16, 17	15, 16, 23
23	16.7, CH_3_	1.29, s	17	17, 18, 19	16, 22, 20
1′	136.8, C				
2′/6′	129.3, CH	7.17, br d (7.7)	3′/5′	10, 2′/6′, 4′	10a, 10b, 3′/5′, 4
3′/5′	129.0, CH	7.32, br t, (7.7)	2′/6′, 4′	1′, 3′/5′	2′/6′
4′	127.1, CH	7.23, m	3′/5′	2′/6′	

^a^ is the order of signal intensity.

**Table 2 molecules-30-00957-t002:** Cytotoxicities of **1**–**4** tested in mouse fibroblast L929 cells.

Compound	IC_50_ [µM] Mouse Fibroblast L929
New cytochalasin **1**	23
18-dehydroxy-cytochalasin E (**2**)	14
Cytochalasin Z_7_ (**3**)	43
Cytochalasin Z_8_ (**4**)	22
Epothilone B (positive control)	0.0016

## Data Availability

All data generated are in the manuscript or the Appendix A. Raw data (i.e., NMR and MS files) are available upon request from the corresponding author.

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
