# Peer review of "Cytochalasins from the Ash Endophytic Fungus Nemania diffusa DSM 116299"

_molecules, 2025, doi:10.3390/molecules30040957_

Round 1
Reviewer 1 Report
Comments and Suggestions for Authors
The manuscript presents the isolation and characterization of a new cytochalasin alongside six known congeners from the endophytic fungus Nemania diffusa. The structural elucidation was achieved using HRMS and 1D and 2D NMR data. The biological activity of these compounds, specifically their cytotoxic effects on mouse fibroblast L929 cells, was also evaluated. While the study is comprehensive and holds merit for publication, the manuscript requires significant revision to address the following critical issues:
1. The description of the structural elucidation for compound 1 is overly simplistic and lacks sufficient detail. The authors should provide a more thorough interpretation of the MS and NMR spectroscopic data, including high-quality HMBC, COSY, and NOESY correlations, directly within the main text. This will enhance the clarity and robustness of the structural assignment. Regarding the stereochemistry of 1, the configurations of C-1, C-2, C-4, C-5, and C-6 are stated to remain unchanged, which is acceptable without additional explanation. However, the stereochemistry of C-18 requires careful investigation, especially to confirm whether the configuration remains consistent following hydroxylation. The authors should provide explicit evidence or reasoning to support this determination.
2. There appears to be an error in the value of C-23 in Table 1. The authors should carefully recheck this assignment and provide the corrected data.
Author Response
- The description of the structural elucidation for compound 1is overly simplistic and lacks sufficient detail. The authors should provide a more thorough interpretation of the MS and NMR spectroscopic data, including high-quality HMBC, COSY, and NOESY correlations, directly within the main text. This will enhance the clarity and robustness of the structural assignment. Regarding the stereochemistry of 1, the configurations of C-1, C-2, C-4, C-5, and C-6 are stated to remain unchanged, which is acceptable without additional explanation. However, the stereochemistry of C-18 requires careful investigation, especially to confirm whether the configuration remains consistent following hydroxylation. The authors should provide explicit evidence or reasoning to support this determination.
Thank you for this assessment! We extended the description of the structure elucidation of 1 and describe the elucidation of the planar as well the stereoconfiguration in detail. In particular, we added figure 2B to explain the assignment of the configuration of C-18.
- There appears to be an error in the value of C-23 in Table 1. The authors should carefully recheck this assignment and provide the corrected data.
Thanks for your attentive reviewing! We corrected this typo.
Reviewer 2 Report
Comments and Suggestions for Authors
In this work, a new cytochalasin and six known ones were obtained from the endophytic fungus Nemania diffusa. The chemical structure of new compound was established by the extensive spectral data. Moreover, some metabolites displayed different levels of bioactivity. And the analysis of structure-activity relationship was studied. All these findings were important, and this work was suggested to be published in this journal after the following revisions were done.
1. Please check the name of compound 7 in the Abstract and the subsections ‘2.1. Isolation and structure elucidation of the compounds’ and ‘4.6. Spectral data’, which was not the same as that reported in the literature.
2. Was there any cytochalasin reported from the fungus of the genera Nemania? This was important to figure out in the Introduction.
3. The description of protons were wrong. Indeed, there were nine aromatic/olefinic (two with dual intensities) as well as seven aliphatic methines, one methylene and four methyl groups.
4. Please check the NMR data in the Table 1 carefully, because some date were not consistent with those observed in Figures S2 and S3, for instance, H-19 and C-23.
5. The description ‘three aliphatic carbons without bound protons’ was inappropriate. Please clearly figure out the different types of carbons, and the corresponding functionalities, such as epoxide.
6. Please put Figure S22 in the main text. And draw the ROESY correlations in the 3D confirmation structure, not a planar structure.
7. How to determine the configuration of C-23? This should be clearly pointed out, because the chemical shift of this carbon was completely different from other congers. Moreover, did you make your efforts to determine the absolute configuration of compound 1?
8. To help the readers get a better understand, it was important to draw a figure to show the ROESY correlations of compound 2, and provide the 2D NMR spectra in the Supplementary Materials.
9. Please change the unit ‘μg/mL’ to ‘μM’, because different compounds had different molecular weights. Then started the analysis of structure-activity relationship.
10. It would be better to assign ‘Molecular analysis’ as a new subsection.
11. Please check the ‘4.6. Spectral data’ carefully, because some data were not consistent with those observed in the Supplementary Materials, for instant, the HRESIMS data of compound 7.
12. The styles of references 2 and 3 were different from others. And provide the page or article number of reference 8.
13. Some sentences were difficult to understand, such as ‘Notably, we discovered unexpectedly a high number of irreversible compounds (1, 2, and 4) and were able to emphasize structural features…’ and ‘Structurally closest known relative of 1 is 2 with 1 being the 18-hydroxyderivative of 2.’.
14. Please pay attentions to the Italic font for the species name ‘N. diffusa’(P2L60) and ‘Nemania diffusa’(Figure 1 caption), and the subscript font of the number in ‘H2O’(P8L273). Please check other typo errors.
15. Table 1 caption: ‘CDCl3-d’ → ‘CDCl3’
Author Response
1.Please check the name of compound 7 in the Abstract and the subsections ‘2.1. Isolation and structure elucidation of the compounds’ and ‘4.6. Spectral data’, which was not the same as that reported in the literature.
Compound #7 has been described first by Steyn under the name “cytochalasin K”. Unfortunately, another cytochalasin was described later under the very same name. Thus, we use the term cytochalasin K Steyn to be specific, in the same way as Zhu et al. 2021 (https://doi.org/10.1007/978-3-030-59444-2_1).
- Was there any cytochalasin reported from the fungus of the genera Nemania? This was important to figure out in the Introduction.
There is a single publication describing the isolation of epoxy-cytochalasins from Nemania sp. UM10M. We had cited this paper in the discussions part. Due to the remark of the reviewer we mention it in the introduction, too. Furthermore, we added two citations to highlight the secondary metabolite potential of Nemania in general.
- The description of protons were wrong. Indeed, there were nine aromatic/olefinic (two with dual intensities) as well as seven aliphatic methines, one methylene and four methyl groups.
Corrected
- Please check the NMR data in the Table 1carefully, because some date were not consistent with those observed in Figures S2and S3, for instance, H-19 and C-23.
Both typos were corrected.
- The description ‘three aliphatic carbons without bound protons’ was inappropriate. Please clearly figure out the different types of carbons, and the corresponding functionalities, such as epoxide.
We rephrased the described to “three oxygen-binding, aliphatic carbons without bound protons”, since it is not possible to be more specific just with the chemical shift data.
- Please put Figure S22in the main text. And draw the ROESY correlations in the 3D confirmation structure, not a planar structure.
Done as suggested.
- How to determine the configuration of C-23? This should be clearly pointed out, because the chemical shift of this carbon was completely different from other congers. Moreover, did you make your efforts to determine the absolute configuration of compound 1?
We added the explanation of the C–17 and C–18 (carbon carrying methyl C–23) in detail.
- To help the readers get a better understand, it was important to draw a figure to show the ROESY correlations of compound 2, and provide the 2D NMR spectra in the Supplementary Materials.
We added figure 2B as a visual presentation for ROESY correlations as suggested.
- Please change the unit ‘μg/mL’ to ‘μM’, because different compounds had different molecular weights. Then started the analysis of structure-activity relationship.
We calculated the IC50 values in µM and replaced the “µg/mL” values in table 2. The structure-activity relationship does not change since the molecular weights are similar.
- It would be better to assign ‘Molecular analysis’ as a new subsection.
The changes have been made in a different yet clearer manner.
- Please check the ‘4.6. Spectral data’ carefully, because some data were not consistent with those observed in the Supplementary Materials, for instant, the HRESIMS data of compound 7.
All the spectra have been checked and necessary changes have been made.
- The styles of references 2 and 3 were different from others. And provide the page or article number of reference 8.
The changes have been made accordingly.
- Some sentences were difficult to understand, such as ‘Notably, we discovered unexpectedly a high number of irreversible compounds (1, 2, and 4) and were able to emphasize structural features…’ and ‘Structurally closest known relative of 1 is 2 with 1 being the 18-hydroxyderivative of 2.’.
The sentences have been modified and restructured in a clearer and more effective manner.
- Please pay attentions to the Italic font for the species name ‘N. diffusa’(P2L60) and ‘Nemania diffusa’(Figure 1caption), and the subscript font of the number in ‘H2O’(P8L273). Please check other typo errors.
The changes have been made accordingly and all the typo errors have been corrected.
- Table 1caption: ‘CDCl3-d’ → ‘CDCl3’
The change has been made accordingly.
Round 2
Reviewer 1 Report
Comments and Suggestions for Authors
The author has addressed all the reviewer's concerns, and the manuscript is now suitable for publication.
Author Response
Thank you very much for your help to improve our manuscript!
Reviewer 2 Report
Comments and Suggestions for Authors
This manuscript had been improved significantly. Only minor revisions were required.
1. The relative configurations of some chiral carbons such as C-3 and C-9 in Figure 2B were not the same as those shown in Figure 1. Please check it carefully.
2. The authors forgot to answer the question ‘Did you make your efforts to determine the absolute configuration of compound 1?’.
3. P9L91: subscript font ‘H3–23’ → ‘H3–23’
Author Response
Thank you very much for your review!
- Some centers were displayed unclear in figure 2B. Thus, we rotated the molecule a bit and redrew the arrows again. Now stereocenters C-3 and C-9 can be viewed easily.
- We did not address this question since the absolute configuration of some basic centers like C-3, C-8 and C-9 is the same for all cytochalasins so far. These centers have been assigned by Xray analysis several times for different compounds.
- Changed as suggested.